# The Role of Molecular Profiling to Predict the Response to Immune Checkpoint Inhibitors in Lung Cancer

**DOI:** 10.3390/cancers11020201

**Published:** 2019-02-10

**Authors:** Courèche Kaderbhaï, Zoé Tharin, François Ghiringhelli

**Affiliations:** 1Department of Medical Oncology, Centre Georges-François Leclerc, 21000 Dijon, France; cgkaderbhai@cgfl.fr (C.K.); z.tharin@gmail.com (Z.T.); 2Department of medicine, University of Burgundy and Franche-Comté, 21000 Dijon, France

**Keywords:** Lung cancer, immunotherapy, biomarkers

## Abstract

Immune checkpoint inhibitors radically changed the treatment of patients with non-small cell lung cancer (NSCLC). However, only one-quarter of patients benefit from these new therapies when used as monotherapy. The assessment of Program Death Ligand-1 (PD-L1) tumor expression by immunohistochemistry is used to select potential responder patients, but this not an optimal marker since it does not predict the absence of anti PD-1 efficacy. Despite this shortcoming, PD-L1 remains the gold standard biomarker in many studies and the only biomarker available for clinicians. In addition to histological markers, transcriptomic and exome analyses have revealed potential biomarkers requiring further confirmation. Recently, tumor mutational burden has emerged as a good surrogate marker of outcome. In this review we will detail current knowledge on DNA and RNA related biomarkers.

## 1. Introduction 

Lung cancer is a major public health issue and the leading cause of cancer-related death worldwide. In 2012, 1.8 million new cases were diagnosed worldwide, representing about 13% of all detected cancers [1]. Currently, it is well established that the immune system plays a key role in both the control of tumor induction, and tumor progression [2,3]. Consequently, the management of non-small cell lung cancer (NSCLC) has taken a major step forward with the development of new therapies targeting immune checkpoints [4]. Nivolumab, Pembrolizumab and Atezolizumab are antibodies directed against Program Death 1 (PD-1) or PD-L1 that disrupt the engagement of PD-1 with its ligands (PD-L1 and 2) and prevent inhibitory signals in T cells. In consequence, cytotoxic T cells are activated and can play their role against tumor cells. A few years ago, the standard of care for patients with no targetable oncogenic drivers was a platinum-based drug doublet [5,6,7]. The arrival of these anti PD-1/PD-L1 monoclonal antibodies has changed the first line of treatment of these patients with the possibility of using Pembrolizumab when tumor cells have a PD-L1 expression of at least 50% [8]. For patients who have a PD-L1 tumor cell expression under 50%, the association of anti PD-1/PD-L1 monoclonal antibodies with chemotherapy has shown to be superior to chemotherapy alone [9,10]. Concerning the second line of treatment, the three antibodies described above can be administrated with better response and survival rates than standard chemotherapy. However, only 20% to 25% of NSCLC patients experience a sustainable response to immune-checkpoint inhibitors. Therefore, finding biological or clinical biomarkers that could help select patients who will respond to immune checkpoint blockades remains a key challenge (Table 1).

## 2. PD-L1 Expression, the First but Imperfect Biomarker 

PD-L1 expression is the first biomarker routinely used for patients treated with immunotherapy. Its efficacy as a biomarker has been evaluated in many clinical trials testing immunotherapy in NSCLC. Expression of PD-L1 is determined by the proportion of tumor cells and/or immune cells that express PD-L1 by immunochemistry. Currently, four antibody clones (22C3, 28-8, SP263 and SP142) are approved to measure PD-L1 expression in patients with NSCLC. Generally, high PD-L1 expression is associated with higher response rates to anti PD-1/PD-L1 therapies and consequently clinical benefits [8,16]. However, some patients with high expression of PD-L1 will not respond to anti PD-1/PD-L1 therapies. The opposite is also true with a proportion of patients with negative PD-L1 expression who will benefit from these treatments [17,18,19].

Many reasons can explain this shortcoming. Firstly, various antibody clones with different cutoff values and scoring evaluations were used in different clinical trials [20]. Therefore, it is impossible to compare different trials results. For example, the OAK study that tested Atezolizumab in patients with advanced or metastatic NSCLC previously treated by chemotherapy used a composite score evaluating the expression of PD-L1 by tumor cells and tumor microenvironment immune cells, whereas the clinical trials evaluating the efficacy of Nivolumab or Pembrolizumab in the same indication evaluated the expression of PD-L1 only in tumor cells. Secondly, PD-L1 expression suffers from important temporal heterogeneity. Indeed, PD-L1 expression can be influenced by the routinely used oncologic treatments. For example, chemotherapy and/or radiotherapy can induce immunogenic cell death and the production of Type I or type II interferon [21,22] or cause intracellular oncogenic variations (loss of PTEN, amplification au JAK2, production of HIF-1a) [23,24,25]. In consequence, it is important to evaluate PD-L1 expression on recent biopsies. Finally, PD-L1 expression has a spatial heterogeneity. Indeed PD-L1 expression can be different from one metastatic site to another in patients with NSCLC. [26]. Moreover, it can vary within different areas in the same tumor location. This was brought to light when discordances were found in PD-L1 expression between biopsy specimens and surgical resections of the same tissue. For example, Ilie et al. compared PD-L1 expression between preoperative biopsies and the corresponding lung resection in 160 patients presenting an NSCLC. They found major discordances in PD-L1 expression between the biopsies and the tumor samples [27].

Thereby, PD-L1 expression is an interesting biomarker to try to select patients who might be prone to respond to anti PD-1/PD-L1 therapies. However, this biomarker is imperfect, especially nowadays that treatment combinations (chemotherapy + anti-PD-L1/PD-1 antibody), are an option as a first line of treatment for advanced or metastatic NSCLC patients. Therefore, it is essential to find a more specific and sensitive biomarker to select patients who are more likely to respond to chemotherapy alone, anti-PD-L1/PD-1 antibody alone or combotherapy.

## 3. Tumor Mutational Burden and Response to Immunotherapy

It is well known that many mechanisms can lead to the accumulation of somatic mutations in the cancer genome [28]. Lung tumors are one of the types of cancer that harbor the most of synonymous and non-synonymous genetic alterations. It has recently been proved that the accumulation of mutations in tumor cells enhances the immunogenicity of cancer cells [29,30]. Indeed, these genomic alterations code for mutated proteins. In cancer cells, these mutant proteins can be cleaved by the proteasome into neopeptides, which will be presented to Human Leukocyte Antigen (HLA) molecules expressed by malignant cells. In other cases, dendritic cells will phagocytize dying tumor cells then present neopeptides to trigger an antitumor T cell immune response. Consequently, elicited T cell are recruited in the tumor microenvironment and exert an antitumoral immune response and therefor limit tumor growth (Figure 1). PD-1 expression on T-cell surface, leads to quiescence and exhaustion. These tumor infiltrating lymphocytes (TILs) can be reactivated by anti PD-1/PD-L1 antibodies leading to an antitumoral effect [31]. The total number of mutations (synonymous and non-synonymous) in tumor cells is called tumor mutational burden (TMB). The level is evaluated by the number of mutations per megabase based on new generation sequencing (NGS) technologies using whole exome sequencing (WES) or large NGS panels. Several recent studies have highlighted a link between TMB and response to anti PD-1/PD-L1 treatments. Indeed, Yarchoan et al. found a correlation between TMB and objective response rates to immunotherapy in a cohort, including 27 different tumor types [32]. In lung cancer, TMB has been studied as a predictive biomarker of response to anti PD-1/PD-L1 therapy after having noticed that smoker history was associated with higher response rates to immunotherapy and to high TMB. Tobacco is a mutagen agent which induces DNA damage and high levels of mutation [33]. Rivzi et al. showed in 2015 in two independent cohorts of NSCLC treated with anti PD-1 antibody (Pembrolizumab) that higher non-synonymous TMB was associated with better objective response rates and better progression-free survival (PFS) under Pembrolizumab [34]. Another study led by Spigel et al. confirmed that high non-synonymous TMB were associated with better objective response rates and better progression-free survival (PFS) in NSCLC treated with anti PD-1 antibody or PD-L1. Patients with high TMB were treated for longer than those with low TMB with a cut-off of 15 mutations per megabase (64 weeks vs. 17 weeks, *p* = 0.010) [35]. Additional data, extracted from an exploratory analysis of the CheckMate 026 study, brings interesting results concerning TMB as an independent predictive factor. This study compared chemotherapy to Nivolumab in patients with previously untreated stage IV or recurrent NSCLC with a PD-L1 expression level of at least 5% [36]. This exploratory analysis was conducted on 58% of the patients who had undergone randomization. TMB was determined by whole exome sequencing. Patients with high TMB had a higher response rate (47% vs. 28%) and the PFS was longer (9.7 vs. 5.8 months) in the Nivolumab group. The selected cutoff was of 243 mutations which correspond to about five mutations per megabase. Conversely, the use of Nivolumab seems to be deleterious for patients with low TMB with a shorter PFS than in the chemotherapy group [36,37]. To sum up, several retrospective analysis or studies have brought to light strong evidence of the predictive impact of TMB in the response to anti PD-1/PD-L1 immunotherapy in patients with NSCLC [38,39]. However, to confirm this new paradigm, prospective studies are mandatory. The phase III study CheckMate 227 prospectively analyzed the response to immunotherapy depending on TMB in patients with stage IV NSCLC. In this first line strategy study, patients with chemotherapy-naïve stage IV or recurrent NSCL and with ≥1% PD-L1 expression were randomly assigned to receive either standard chemotherapy, or Nivolumab + Ipilimumab, or Nivolumab alone. Patients with negative PD-L1 expression were also randomized between standard chemotherapy, Nivolumab + Ipilimumab or Nivolumab + chemotherapy [11]. Based on ancillary analysis of CheckMate 568, a phase II trial evaluating Nivolumab + Ipilimumab the protocol was modified to randomize patients in function of TMB. Cut-off of at least 10 mutations per megabase was chosen to select patients who are more likely to respond to this double immunotherapy, independently of PD-L1 expression [40]. In the CheckMate 227 study, the 1-year PFS is higher in the “Nivolumab + Ipilimumab” arm versus the chemotherapy group (42.6% vs. 13.2%; HR 0.58, 95% CI: 0.41–0.81; *p* < 0.001) for patients with high TMB. For patients with low TMB, the results are similar (HR 1.07, 95% CI: 0.84–1.35). Updated data presented at ESMO 2018 from CheckMate 227, showed that the median overall survival (OS) for the “Nivolumab + Ipilimumab” arm for patients with TMB ≥ 10 mut/Mb was of 23.03 months compared to 16.72 months for the chemotherapy arm (0.77; 95% CI: 0.56–1.06). Among patients with TMB < 10 mut/Mb, the median OS was of 16.20 months vs. 12.42 months, respectively (HR 0.78; 95% CI: 0.61–1.00). These results confirm that TMB is an interesting tool as a predictive factor of response to immunotherapy and of PFS in NSCLC. Moreover, it has been shown that patients with high TMB benefit from a double immunotherapy independently of PD-L1 expression or histology. Importantly, TMB is not correlated to PD-L1 expression, suggesting that both variables could be complementary. However, OS data from Checkmate 227 suggest that TMB is also a prognostic factor, suggesting caution on its use in patient selection for treatment with a combination of Nivolumab with Ipilimumab. The prognostic role of TMB was confirmed in resected NSCLC where high nonsynonymous TMB (>8 mutations/Mb) was prognostic of favorable outcome [41] (Figure 1).

Surprisingly, opposed to Checkmate 026, Checkmate 227 TMB seems to be a predictive factor for the efficacy of double immunotherapy only (association of anti PD-1/PD-L1 and anti CTLA-4). In a secondary endpoint, the efficacy of Nivolumab (71 patients) versus chemotherapy (79 patients) among patients with a tumor mutational burden of at least 13 mutations per megabase and a PD-L1 expression level of at least 1% was tested. No significant difference was observed between Nivolumab alone and chemotherapy for patients with high TMB (HR 0.95, 97.5% CI: 0.61–1.48; *p* = 0.78) [11].

Concerning anti PD-L1 mAb Atezolizumab, prognostic role of TMB was evaluated in the POPLAR phase II study and the phase III OAK study. In these randomized trials Atezolizumab was superior to docetaxel in the second line of treatment for NSCLC. In the phase III study, OS was of 13.8 months in the Atezolizumab arm versus 9.6 months in the docetaxel arm (ratio (HR 0.73, 95% CI: 0.62–0.87; *p* = 0.0003)) [15,17]. In these 2 studies TMB was evaluated using tumor and blood TMB evaluation. Patient’s serum contains cell free tumor DNA that can be analyzed by NGS technology. Blood draw has distinct advantages compared to tissue biopsy collection. Indeed, blood offers a ready, contemporaneous source of diagnostic material and such testing is less susceptible to potential sampling biases that are associated with single-site tissue biopsies [42,43]. This strategy is feasible and a positive correlation between tumor TMB and blood TMB was observed (Spearman rank correlation = 0.64; 95% CI: 0.56–0.71). Using a cut-off point of 16 mutations per Megabase, the PFS HR was 0.57 (95% CI: 0.33–0.99) for patients with high blood TMB. The median PFS was 4.2 months in the Atezolizumab arm and 2.9 months in the docetaxel arm; the median OS values were 13.0 and 7.4 months, respectively. Importantly blood TMB is independent of the PD-L1 expression measured by IHC. Patients with high blood TMB and high expression of PD-L1 (TC3 or IC3) have the best clinical benefit from Atezolizumab (PFS HR 0.38, 95% CI: 0.17–0.85; OS HR 0.23, 95% CI: 0.09–0.58).

Together, these data support that TMB could be used to predict prognosis of patients with metastatic NSLCLC. In the second line of treatment, TMB is associated with better response and better survival rates when anti PD-1 or PD-L1 mAb are used in comparison to chemotherapy. In the first line of treatment, high TMB is associated with better response rates when patients are treated with a combination of Nivolumab and Ipilimumab in comparison to chemotherapy (Figure 2). As for anti PD-1 used in monotherapy, conflicting results are observed between Checkmate 227 and Checkmate 026 clinical trials and additional studies will be necessary to obtain a clear conclusion Based on current literature PD-L1 > 50% of staining will be used to propose anti PD-1 mAb monotherapy in first line. For other patients in case of high TMB patients will be treated with combo-immunotherapy using anti PD-1 and anti CTLA-4. The other patients will receive chemotherapy with anti PD-1/PD-L1 therapy. Addition works are requested to determine if some patients with no PD-L1 expression and low TMB did not benefit from checkpoint inhibitor and will receive only chemotherapy.

## 4. Limitations of TMB as a Predictive Factor

TMB appears to be a surrogate marker in the prediction of response to checkpoint inhibitors and more precisely for anti PD-1/PD-L1 and anti-CTLA4 therapy. However, further research is necessary concerning some limitations. For example, TMB is a dynamic biomarker with spatial and temporal heterogeneity and there is an important distinction between clonal mutations (carried by all tumor cells) and subclonal mutations (expressed by a fraction of tumor cells). When a tumor is subclonal some clones can potentially have low TMB and little neoantigens and could therefore be ignored by the immune system. Indeed, it has been shown that homogeneous tumors with monoclonal composition and high TMB experience better response to anti PD-1/PD-L1 antibodies in comparison to subclonal heterogenous tumor [44]. However, TMB determination is performed on a small biopsy which limits the evaluation of spatial heterogeneity. TMB determination on blood samples could help with that matter. As said before, TMB also presents temporal variability. After injecting immunotherapy, TMB could change. For example, in the setting of melanoma, reduction of TMB is associated with response to checkpoint inhibitors [45]. Further works are warranted to determine whether first line chemotherapy or target therapies also affect TMB. Another limitation is linked to the reproducibility of the technique. It is necessary to homogenize sequencing technics and mainly gene panels and bioinformatic pipelines in order to use routinely this biomarker in the decision making of treatment for patients with advanced NSCLC. Finally, the cost of this technique is another limitation in the current area where health economics are a primordial issue.

## 5. Other Molecular Predictive Factors

Advanced NSCLC patients with targetable molecular alterations like EGFR activating mutations or ALK rearrangements experience fewer good responses to checkpoint inhibitors. Retrospective analysis, carried out on patients with NSCLC advanced adenocarcinoma with EGFR mutations or ALK rearrangements and treated with anti-PD-L1 antibodies, showed low response rates (5% and 0% response rates respectively) [46]. Two recent meta-analysis confirmed these results for patients with EGFR mutations. The first one collected survival data from three randomized phase III studies comparing Docetaxel and anti PD-1 antibody (Nivolumab or Pembrolizumab) as second line of treatment. No benefit in OS was found for checkpoint inhibitors for these patients (*n* = 186, HR 1.05; *p* < 0.81) [47]. Other meta-analysis collected survival data from 5 studies comparing Docetaxel to Nivolumab/Pembrolizumab or Atezolizumab in the second line of treatment for NSCLC. Again, no benefit in terms of survival were found with checkpoint inhibitors for patients with an EGFR mutation (*n* = 271, HR 1.11; *p* = 0.54) [48]. These findings suggest that checkpoint inhibitors should not be an option as first line of treatment for these patients and should only be considered after having exhausted all available targeted therapies.

Other driver mutations were recently found to be associated with a response to anti PD-1/PD-L1 mAb. STK11, EGFR, KRAS and P53 are the most frequently mutated genes found in NSCLC. These mutations are associated with local immune reactions. The presence of *TP53* mutations without co-occurring *STK11* or *EGFR* led to the identification of a group of tumors with high CD8 infiltration and PD-L1 expression. Higher PFS was observed for patients treated with anti PD-1 if harboring *TP53*-mut/*STK11*-*EGFR*-WT tumors. (PFS: HR 0.32; 95% CI: 0.16–0.63, *p* < 0.001) [12,30]. STK11 mutations are associated with decrease expression of STING, a cytosolic DNA sensor. This event reduces Interferon (IFN) type production and reduce CD8 recruitment. Such results underline that immune effect of mutation is probably more important than their driver effect to predict response to immunotherapy [49] Recently, our group found that in addition to TMB, a high number of neoantigens, mutational signatures 1A and 1B, mutations in DNA repair pathways and a low number of TCR clones are associated with better PFS. Combination of these variables outperformed TMB and PD-L1 as a predictive biomarker [50] (Figure 2).

Mismatch-repair deficiency is a DNA repair enzyme deficiency, which induce high number of mutations in a tumor. In the setting of colorectal cancer mismatch repair–deficient tumors have 10 to 100 times as many somatic mutations as mismatch repair–proficient. Moreover, mismatch repair–deficient cancers are richely invaded by lymphocyte. Such tumors were reported to have a high response rate with around 50% of objective response with anti PD-1 mAb [13] MHC-I complex is required for cancer neoantigens presentation to CD8^+^ cells. Genetic mutations code for mutant proteins that are processed into mutant peptides. Variability among the genes encoding for *HLA-I* genes (*HLA-A*, *HLA-B*, and *HLA-C*) and *B2M*, which is required for HLA-I function, has been demonstrated to negatively influence checkpoint inhibitors efficacy. The truncation of *B2M* has been described as a mechanism of resistance in melanomas treated with checkpoint inhibitors [51]. More generally, *HLA* loss of heterozygous alterations are associated with lower survival rates among cancer patients treated with checkpoint inhibitors [52]. Conversely, increased heterozygosity at *HLA-I* loci and expression of certain HLA subtypes like HLA-B44 are associated with better OS under checkpoint inhibitor therapy [53].

Interestingly a recent report also supports the rational that genomic biomarkers could be associated with hyperprogression induced by checkpoint inhibitors. Hyperprogression is defined by an acceleration of tumor progression ranging from 4% to 29% across multiple histologies [54]. In a recent series of patients treated with anti PD-1/PD-L1 the presence of amplification of MDM2/MDM4, EGFR and DNMT3A mutation were associated with hyperprogression [55].

## 6. Transcriptomic Signature

In addition to DNA markers some RNA markers were suggested to be associated with response to checkpoint inhibitors. Immune infiltration by CD8 T cells is associated with response to immunotherapy. A recent study by Tumeh et al. found that preexisting CD8^+^ T cells are essential for tumor regression following therapeutic PD-1/PD-L1 blockade in metastatic melanoma, indicating that CD8^+^ TILs play a key role in anti PD-1 therapy response [56]. Similar observations were done by our group in the second line of treatment of NSCLC cancer patients treated with Nivolumab, generalizing the observation. Based on these studies, some other authors put forward a tumor classification according to T cell infiltration and PDL1 expression [57]. Four categories were defined: Type I adaptive immune resistance (PD-L1 positive and high TILs), type II immune ignorance (PD-L1 negative and low TILs), type III intrinsic induction (PD-L1 positive and low TILs), and type IV immune tolerance (PD-L1 negative and high TILs) pathways.

Transcriptomic analysis could be a valuable strategy to determine immune infiltration in the tumor, but also to assess immune cells function. Using analysis of tumor whole transcriptome, some groups isolated a transcriptional signature called Immunologic Constant of Rejection (ICR) [58]. The ICR signature is currently represented by twenty transcripts and four functional categories: CXCR3/CCR5 chemokines (including *CXCL9*, *CXCL10*, *CCL5*), Th1 signaling (including *IFNG*, *IL12B*, *TBX21*, *CD8A*, *STAT1*, *IRF1*, *CD8B*), effector (including *GNLY*, *PRF1*, *GZMA*, *GZMB*, *GZMH*) and immune regulatory (including *CD274*, *CTLA4*, *FOXP3*, *IDO1*, *PDCD1*) functions. The ICR could be used as a positive predictive biomarker of responsiveness to immunotherapy and as a favorable prognostic marker for various tumor types [14,59,60,61]. This observation suggests that transcriptomic analysis could be used to determine immune infiltration and that this variable is associated with response to immunotherapy [59,62]. This observation was further validated in other clinical trials [14,63,64]. Specifically, Ayers et al. define an interferon signature and an extended immune signature in patients treated with Pembrolizumab [55]. This signature is associated with response rate and survival in patients treated with anti PD-1. In the setting of lung cancer, we tested a simplified immune signature using CD8A and CD274 mRNA quantification using RNAseq. We observed that this signature was predictive of response to Nivolumab. We have analyzed a cohort of 85 patients treated with Nivolumab in the second line or beyond and observed that high CD8 mRNA expression was significantly associated with response rate and progression free survival. Similarly high *CD274* expression was associated with better progression free survival. When combining these dichotomous markers, we observed that patients with high *CD8A* and *CD274* coexpression had longer progression free survival. Interestingly, the combination of the two factors outperformed the discriminatory properties of CD8A or CD274 variables alone, as well as CD8^+^ TILs and PD-L1 IHC variables. In a multivariate model, this biomarker remains independently associated with progression free survival and overall survival. Additionally, these observations were also validated in a cohort of 44 patients [65]. Moreover, we observed that the CD8A/CD274 two-genes signature was superior to two previously published gold-standard signatures [62].

Transcriptional signatures were investigated for Atezolizumab treatment of NSCLC. First line association of chemotherapy plus bevacizumab with or without Atezolizumab was tested in a phase III trial [66]. A T effector gene signature was defined as the expression of *PD-L1*, *CXCL9*, and *IFN-γ* messenger RNA, determined using RNA isolated from macrodissected tumor tissue, obtained at baseline and measured with a dedicated real-time quantitative polymerase-chain-reaction assay. In the T effector -high WT population, PFS was significantly longer in the Atezolizumab group than in the chemotherapy group (median, 11.3 months vs. 6.8 months; stratified HR 0.51; 95% CI: 0.38 to 0.68; *p* < 0.001). In the group of patients with low expression of a T effector gene signature, authors did not observe any difference in term of progression free survival between patients treated with chemotherapy or chemotherapy plus Atezolizumab (median, 7.3 months vs. 7.0 months; unstratified HR 0.76; 95% CI: 0.60 to 0.96).

The predictive role of the transcriptomic signature was also evaluated in the second line by the POPLAR study. In this phase II study, authors randomized patients with NSCLC to Docetaxel or monotherapy Atezolizumab group. The T-effector and IFN-γ gene signature was defined by *CD8A, GZMA, GZMB, IFN*γ, *EOMES, CXCL9, CXCL10*, and *TBX21.* T-effector and IFN-γ-associated gene signatures were correlated with PD-L1 expression. Atezolizumab improved OS in patients with tumors characterized by high expression of T-effector-associated and IFN-γ-associated genes (HR 0.43; 95% CI: 0.24–0.77). In contrast Atezolizumab did not improve patient survival in patients with low expression of T-effector-associated and IFN-γ-associated genes [15].

Recent work revealed two distinct mechanisms of tumor immune evasion [2]. Some tumors have a high level of infiltration by cytotoxic T cells, but these T cells tend to be in a dysfunctional state. In other tumors, immunosuppressive factors may exclude T cells from infiltrating tumors [67]. Recently Jiang et al. developed a computational framework called Tumor Immune Dysfunction and Exclusion (TIDE), to identify factors that underlie these two mechanisms of tumor immune escape. TIDE, integrates the expression signatures of T cell dysfunction and T cell exclusion to model tumor immune evasion. The TIDE signatures were tested in melanoma, lung cancer, head and neck cancer, and could predict response rate and PFS in patients treated with anti CTLA-4 and anti PD-1/PDL1. ROC curves were also used to measure the discriminative properties of the TIDE prediction scores. Compared to classically used biomarkers to predict response to checkpoint inhibitors, like PD-L1 expression or transcriptomic signatures like IFN-γ response [14,68], the TIDE signature achieved consistently better performance for both anti-PD1/PD-L1 and anti-CTLA4 therapies [69].

A novel strategy based on transcriptomic signatures is emerging as a promising approach to defining biomarkers of response to checkpoint inhibitors. Signatures targeting IFNγ and immune infiltrates are strongly correlated with response, PFS and OS in patients treated for lung cancer in the first line or further. These novel biomarkers could prove to be a useful tool, but additional studies are necessary to validate the optimal gene list and technical approach, as well as a valid cut off for patient selection.

New sequencing technology could be used to test T cell repertoire with TCR sequencing. This technic could be used to test expansion and diversity of T cell clones found in the tumor bed. CTLA-4 blockade resulted in both expansion and loss of T cell clonotypes. Improved overall survival was associated with maintenance of high-frequency clones at baseline. In contrast, decrease-diversity of clonotype with treatment is associated with short overall survival [70]. Opposite results were observed with anti PD-1 [45] thus suggesting that anti CTLA-4 and anti PD-1 mAb may act by a different molecular mechanism.

## 7. Microbiota

Some recent studies have highlighted that the composition of the gut microbiota can influence response to immune checkpoint inhibitor. Seminal observations from Laurence Zitvogel group underline in mouse preclinical models that checkpoint inhibitors lose their antitumor efficacy in germ free mice or mice treated with large spectrum antibiotics [71]. Interestingly oral supplementation of favorable microbiota could restore the therapeutic efficacy of checkpoint blockade in mice models. In human also it was observed in NSCLC, Clear Cell Renal carcinoma and melanoma that antibiotics usage before or during anti PD-1/PD-L1 mAb impede their efficacy and such patients showed significantly shorter survival [72]. Even though different microbial species may be associated with response to checkpoint inhibitor, there seems to be a trend toward an enrichment in *Firmicutes* and *Akermansia* in responders and *Bacteroidetes* in non-responders.

## 8. Conclusions

Checkpoint inhibitors have radically changed the treatment of lung cancer. These treatments have demonstrated their efficacy in the first and the second line of treatment of patients suffering from NSCLC and drastically changed the prognosis of this pathology. Despite these results only a subset of patients responds to these treatments. Tumor mutational burden became a major biomarker to predict treatment efficacy. Further work will be required to validate optimal cutoff and standardized methodology. Additional genetic biomarkers are emerging like neoantigens, HLA typing, DNA damage repair anomalies. Such biomarkers are dichotomic and will rapidly implement the Tumor mutational burden. In addition, transcriptomic signatures are also emerging and demonstrate the capacity to predict response to checkpoint inhibitor in phase III trials. These signature estimate T cell infiltration, but also could study T cell effector function, T cell exhaustion and immune—exclusion. Current limitations are based on the absence of standardization and cutoff. Comparative studies of histological analysis and transcriptomic signatures are required to determine whether transcriptomic signatures outperform immunohistological biomarkers. Such work is essential for patient care and to fight financial toxicity. Such results will quickly lead to precision medicine in immuno-oncology.

## Figures and Tables

**Figure 1 cancers-11-00201-f001:**
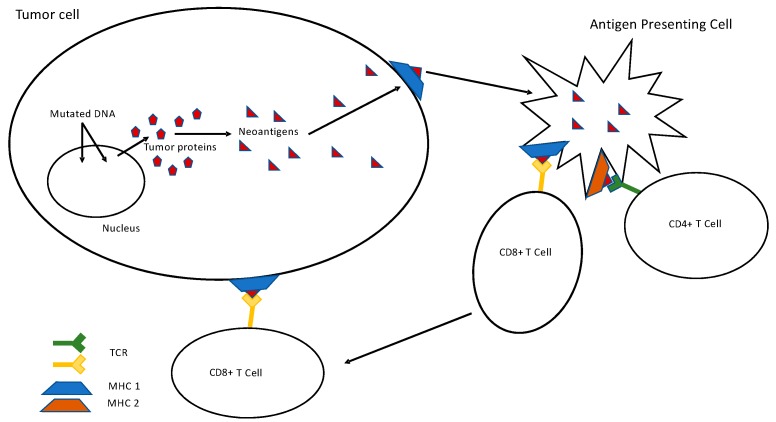
Link between Tumor Mutational Burden and T specific antitumoral response. Abbreviations: DNA, Deoxyribonucleic Acid; MHC, Major Histocompatibility Complex; TCR, T-cell Receptor.

**Figure 2 cancers-11-00201-f002:**
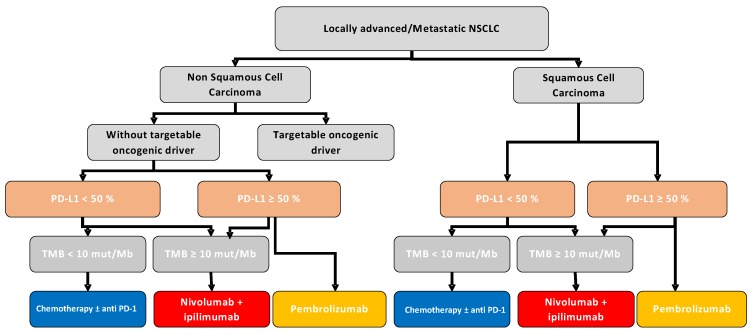
Involvement of TMB in the first line of NSCLC management strategy. Abbreviations: NSCLC, Non-Small-Cell Lung Cancer; Mut, Mutations; Mb, Megabase; PD-L1, Program Death-Ligand 1; TMB, Tumor Mutational Burden.

**Table 1 cancers-11-00201-t001:** Summary of the different genomic and transcriptomic biomarkers available for clinician.

Type		Clinic Usage	Reference
Genomic	TMB	>10 mutation/MB Is associated with better efficacy of nivolumab plus ipilimumab combotherapy	[11]
	STK11/EGFR mutation	Mutation associated with absence of efficacy of anti PD-1	[12]
	Mismatch repair deficiency	50% response rate whatever the tumor type (with anti PD-1 mAb)	[13]
Transcriptomic	IFN signature	High expression is associated with better efficacy of anti PD-1	[14]
	Extended Immune signature	High expression is associated with better efficacy of anti PD-1	[15]

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
