# Peer review of "The Role of Molecular Profiling to Predict the Response to Immune Checkpoint Inhibitors in Lung Cancer"

_cancers, 2019, doi:10.3390/cancers11020201_

Reviewer 1 Report

This is a timely review that has nicely summarized key findings in this topic. While I find this article potentially publishable, there are several issues need to be address beforehand.

A. Major issues:

- The authors need to include the value of microbiome as a predictor

- Need to mention the markers that are relevant/predictive to "hyper-progression", which is important so we do not necessarily risk our patients.

- The authors need to include the discussion of Figure 2, e.g. when to consider? how to incorporate into current practice? for example, what should be the biomarker(s) to guide the combination of chemotherapy plus immunotherapy?

- Need to elaborate more regarding the impact of mutations. It does not need to be the driver mutation, for example, STK11 (or LKB1) has emerged as an independent predictor for the resistance to anti-pd1/pd-l1. Interesting mutations also include KRAS, BRAF, etc.

- Paragraph from line 236 to 246 seems to be irrelevant to the "transcriptomic signature".

- It may sound a little deviate from the topic (I will therefore leave this to the authors), but it might be interesting to include a small paragraph regarding what can be done to predict the toxicities from immunotherapy as some of them can be lethal to lung cancer patients, for example, the pneumonitis.

B. Minor issues:

- line 89: "micro environment" needs to be changed to "microenvironment"

- line 118: change "NSCLC-(33-34)" to "NSCLC (33-34)"

- In Figure 1, consider change "CD4" to "CD4+", and "CD8" to "CD8+"

- line 154, "Concerning anti PD-L1 mab Atezolizumab, TMB role of was evaluated.." needs to be rephrased.

- line 158, what dose "(12 et 38)" mean?

- line 183, "iz" meant to be "is"?

- line 193, what does "contraction of TMB" mean? please clarify.

- line 258, "define and interferon signature" suppose to be "define an interferon signature"?

Author Response

A. Major issues:

- The authors need to include the value of microbiome as a predictor:

RESPONSE: accordingly we add a paragraph on microbiota

- Need to mention the markers that are relevant/predictive to "hyper-progression", which is important so we do not necessarily risk our patients.

RESPONSE: Few data are currently avaible on biomarker associated with hyper progression. We add some sentence on recent data on this subject

- The authors need to include the discussion of Figure 2, e.g. when to consider? how to incorporate into current practice? for example, what should be the biomarker(s) to guide the combination of chemotherapy plus immunotherapy?

RESPONSE we add a sentence to explain how biomarker will guide therapy

- Need to elaborate more regarding the impact of mutations. It does not need to be the driver mutation, for example, STK11 (or LKB1) has emerged as an independent predictor for the resistance to anti-pd1/pd-l1. Interesting mutations also include KRAS, BRAF, etc.

RESPONSE: We add a sentence on the fact that immune effect of mutation is probably more important than their driver property

- Paragraph from line 236 to 246 seems to be irrelevant to the "transcriptomic signature".

RESPONSE: This paragraph was added to introduce the interest to assess immune infiltrate.

- It may sound a little deviate from the topic (I will therefore leave this to the authors), but it might be interesting to include a small paragraph regarding what can be done to predict the toxicities from immunotherapy as some of them can be lethal to lung cancer patients, for example, the pneumonitis.

RESPONSE: This topic is not directly in the scope of the manuscript so we do not add modification on the subject

B. Minor issues:

RESPONSE: we make the appropriate modifications

- line 89: "micro environment" needs to be changed to "microenvironment"

- line 118: change "NSCLC-(33-34)" to "NSCLC (33-34)"

- In Figure 1, consider change "CD4" to "CD4+", and "CD8" to "CD8+"

- line 154, "Concerning anti PD-L1 mab Atezolizumab, TMB role of was evaluated.." needs to be rephrased.

- line 158, what dose "(12 et 38)" mean?

- line 183, "iz" meant to be "is"?

- line 193, what does "contraction of TMB" mean? please clarify.

- line 258, "define and interferon signature" suppose to be "define an interferon signature"?

Reviewer 2 Report

Overall this is an excellent review which covers many of the key point on the topic under discussion. The authors have limited the study to molecular markers, which removes some of the known non molecular means of assessing response to immunotherapies (flow cytometry assessment of infiltrating immune cells, etc…). I think it is suitable for publication provided a few changes are made.

Major points

1.)    The reviewers must include a discussion of the use of mis-match repair deficiencies, not just TMB, as biomarkers of checkpoint responsiveness. Microsatalite stability as a predictor. See PMID: 26028255.

2.)    The reviewers should discuss immune-sequencing as a means to identify responders.  This technique measures the diversity of the T cell repertoire and uses it to predict responsiveness.  See PMID: 24871131.

Minor issues

Line 98 and Line 118 – inconsistencies using anti-PD-1 vs anti PD-1

Line 126 – Typo TMB.-A

Line 129 and Line 174 - inconsistencies with use of + or and

Line 132 – need space between ESMO and 2018

Line 154 - change mab to mAb

Line 158 – Typo 12 et 38

Line 183 – Typo iz

Line 237 – CD8 is shown and not CD8+ like in the remainder of the paper.

Line 275 Line 280 – Teff is used and not T-effector like the rest of the paper.

Line 285 – docetaxel is used and in other instances it is capitalized

Line 302 vs Line 305 – interferon gamma is spelled out where is other instances it is abbreviated IFNγ

Line 318 – Tumor is capitalized

Author Response

verall this is an excellent review which covers many of the key point on the topic under discussion. The authors have limited the study to molecular markers, which removes some of the known non molecular means of assessing response to immunotherapies (flow cytometry assessment of infiltrating immune cells, etc…). I think it is suitable for publication provided a few changes are made.

Major points

)    The reviewers must include a discussion of the use of mis-match repair deficiencies, not just TMB, as biomarkers of checkpoint responsiveness. Microsatalite stability as a predictor. See PMID: 26028255.

RESPONSE : we aggree and add a paragraph on MSI and hypermutative tumors

2.)    The reviewers should discuss immune-sequencing as a means to identify responders.  This technique measures the diversity of the T cell repertoire and uses it to predict responsiveness.  See PMID: 24871131.

 RESPONSE: We quote this paper and add a paragraph on this concept

Minor issues

REPONSE change are made appropriately

Line 98 and Line 118 – inconsistencies using anti-PD-1 vs anti PD-1

Line 126 – Typo TMB.-A

Line 129 and Line 174 - inconsistencies with use of + or and

Line 132 – need space between ESMO and 2018

Line 154 - change mab to mAb

Line 158 – Typo 12 et 38

Line 183 – Typo iz

Line 237 – CD8 is shown and not CD8+ like in the remainder of the paper.

Line 275 Line 280 – Teff is used and not T-effector like the rest of the paper.

Line 285 – docetaxel is used and in other instances it is capitalized

Line 302 vs Line 305 – interferon gamma is spelled out where is other instances it is abbreviated IFNγ

Line 318 – Tumor is capitalized

Reviewer 3 Report

I would like to thank you of authors and it was really helpful and comprehensive review based on the immune checkpoint inhibitor. The only suggestion that I have please add a table to your review and compare the different biomarkers available for clinicians with molecular profiling for prediction the response.

Thanks again and I hope to continue your research with the same direction.

Author Response

The only suggestion that I have please add a table to your review and compare the different biomarkers available for clinicians with molecular profiling for prediction the response.

Response accordingly we add a table on the different available biomarkers

Round  2

Reviewer 1 Report

Addressed my comments. Ready to be published.

Reviewer 2 Report

This manuscript is sutable for oublication.